# Premorbid Comorbidities as Predictors of Outcome in Ischemic Posterior Fossa Stroke: A Retrospective Evaluation Using the Age-Adjusted Charlson Comorbidity Index

**DOI:** 10.3390/brainsci15080892

**Published:** 2025-08-21

**Authors:** Francesca Culaj, Toska Maxhuni, Stefan T. Gerner, Anne Mrochen, Tobias Braun, Priyanka Boettger, Maxime Viard, Hagen B. Huttner, Martin Jünemann, Omar Alhaj Omar

**Affiliations:** 1Department of Neurology, Justus-Liebig-University, 35392 Giessen, Germany; toska.maxhuni@neuro.med.uni-giessen.de (T.M.); stefan.gerner@uk-erlangen.de (S.T.G.); anne.mrochen@neuro.med.uni-giessen.de (A.M.); tobias.braun@neuro.med.uni-giessen.de (T.B.); hagen.huttner@ukdd.de (H.B.H.); martin.juenemann@neuro.med.uni-giessen.de (M.J.); omar.alhajomar@neuro.med.uni-giessen.de (O.A.O.); 2Center of Mind, Brain & Behavior (CMBB), 35032 Marburg, Germany; 3Translational Neuroscience Network Giessen (TNNG), 35392 Giessen, Germany; 4Department of Neurology, Lahn-Dill-Kliniken Wetzlar, 35578 Wetzlar, Germany; 5Department of Cardiology, Angiology and Critical Care Medicine, Justus Liebig University, 35392 Giessen, Germany; priyanka.boettger@googlemail.com; 6Department of Neurology, Kantonsspital Winterthur, 8401 Winterthur, Switzerland; viard.maxime@googlemail.com

**Keywords:** stroke, posterior cranial fossa, Charlson Comorbidity Index (aCCI), modified Rankin Scale (mRS), NIH Stroke Scale (NIHSS), comorbidities

## Abstract

**Background:** Posterior cranial fossa (PCF) infarctions are associated with elevated mortality rates; however, evidence regarding the prognostic value of comorbidity indices in this context remains scarce. This study investigates the association between the age-adjusted Charlson Comorbidity Index (aCCI) and clinical outcomes in patients with PCF infarctions, aiming to evaluate the aCCI as a prognostic indicator. The aCCI is a validated scoring system that quantifies a patient’s burden of chronic diseases, adjusting for age, to estimate overall comorbidity risk. **Methods:** In this retrospective, single-center analysis spanning two years, patient data were assessed to determine correlations between aCCI scores and clinical outcomes at discharge, specifically the modified Rankin Scale (mRS) and National Institutes of Health Stroke Scale (NIHSS). Structural equation modeling (SEM) was employed to elucidate the relationships among these variables. **Results:** SEM demonstrated that higher pre-stroke comorbidity burden, as quantified by the aCCI, was significantly associated with increased stroke severity and poorer functional outcomes at discharge. The aCCI exhibited strong direct effects on both NIHSS (path coefficient: 0.70) and mRS (path coefficient: 1.43). **Conclusions:** Premorbid comorbidities, as measured by the aCCI prior to stroke onset, are significantly correlated with both neurological deficit and functional outcome at discharge in patients with PCF infarctions. These findings underscore the potential utility of the aCCI as a prognostic tool for outcome prediction in this patient cohort.

## 1. Introduction

Posterior circulation ischemic strokes account for approximately 20% of all strokes, with the cerebellum being the most commonly affected region [1]. Ischemic strokes occurring in the posterior cranial fossa are associated with higher mortality rates and poorer functional outcomes compared to supratentorial strokes originating from the vertebrobasilar circulation [2]. Notably, basilar artery occlusion results in significant disability in approximately one-third of affected patients [3,4].

The vertebral arteries (VAs), originating from the subclavian arteries, supply about 20% of intracranial blood flow and merge to form the basilar artery (BA) at the pontomedullary junction. Key branches—including the posterior inferior cerebellar artery (PICA), anterior inferior cerebellar artery (AICA), and superior cerebellar artery (SCA)—supply the cerebellum and parts of the brainstem [5,6].

The Charlson Comorbidity Index (CCI), introduced by Charlson et al. in 1987 [7], is a standardized tool for quantifying comorbidity burden, with scores ranging from 0 to 37 to reflect increasing severity. Higher CCI scores are linked to greater risk of complications, prolonged hospital stays, poorer long-term outcomes in stroke patients, as well as mortality [7,8]. Recent studies have incorporated age as an additional factor into the Charlson Comorbidity Index (CCI) [9].

The age-adjusted Charlson Comorbidity Index (aCCI) extends the original CCI by incorporating additional points based on patient age. This adjustment allows the aCCI to better account for the combined impact of comorbidities and advancing age on clinical prognosis [8,9].

The impact of the CCI on stroke outcomes has been increasingly investigated over the past decade [10,11]. However, most studies have focused on stroke in general, without accounting for stroke localization or its potential influence on prognosis [12,13,14,15]. There is generally limited research specifically addressing the effect of comorbidities on posterior circulation strokes, particularly those confined to the posterior cranial fossa [16,17]. Furthermore, commonly used scores for assessing stroke severity and burden are more accurate for anterior circulation strokes and less reliable for those located in the posterior cranial fossa [18,19].

This study aims to investigate the impact of pre-existing comorbidities, as measured by the aCCI, on stroke severity and functional outcomes in patients with strokes localized to the ischemic posterior cranial fossa. Given the unique anatomical and clinical characteristics of posterior cranial fossa strokes [20], particularly their higher mortality rates and poorer prognoses compared to other strokes, this research seeks to address the gap in the literature concerning how age and comorbidity burden influence recovery in this specific stroke subtype. By focusing on the relationship between aCCI scores, NIHSS, and mRS at discharge, the study aims to enhance understanding of prognostic factors in ischemic posterior fossa stroke patients and inform more tailored clinical management strategies.

## 2. Materials and Methods

### 2.1. Study Design

To our knowledge, no study has investigated the relationship between the age-adjusted CCI and its prognostic impact on stroke in general, or on stroke in the PCF in particular.

This retrospective, monocentric study was conducted at the Department of Neurology at University Hospital Giessen, Germany. Data for this study were collected over a two-year period, from January 2020 to December 2021. The analysis included patients admitted either to the intensive care unit or the stroke unit of the Department of Neurology at University Hospital Giessen. These patients were examined and compared to evaluate relevant clinical characteristics and outcomes.

Only patients with ischemic stroke localized to the posterior cranial fossa were included as part of the inclusion criteria. Patients with hemorrhagic stroke or strokes involving the territory of the posterior cerebral artery were excluded. Further, to ensure the integrity of the study and minimize bias from pre-existing functional impairments, only patients with a pre-index event mRS score of twoor less were included. Patients who underwent endovascular therapy (EVT) were also excluded to maintain a more homogeneous study population.

This study was part of the retrospective Giessen stroke registry (GIST; prospective part at clinicaltrials.gov ID: NCT05295862), which was approved by the local ethics committee of the faculty of medicine (FB11), Justus-Liebig University Giessen (decision reference: AZ 220/21).

### 2.2. Clinical Parameters

Individual parameters of the selected patients were collected through an electronic medical record review. These parameters included demographic data (such as age and gender), length of hospital stay, risk factors, prior therapy with anticoagulation or antiplatelet agents, and clinical status upon admission based on NIHSS [20], as well as mRS [21], premorbidity status based on aCCI [8], lesion location (cerebellum or brainstem), and large vessel occlusion (LVO) identified in angiography that was not treated with EVT. The aCCI is a validated tool that quantifies comorbidity by assigning weighted scores (1–6) to 19 chronic conditions. The total score, which also incorporates age-related points, reflects overall comorbidity burden; higher scores are associated with increased mortality risk and poorer outcomes

### 2.3. Follow-Up Parameters

Patient outcomes were assessed at discharge using the NIHSS and the mRS. A follow-up after inpatient treatment could not be conducted due to incomplete data. The outpatient follow-up data were incomplete and not assessed for every patient, rendering them ineligible for analysis.

### 2.4. Outcome Measures

The primary endpoint of this study was to evaluate the hypothetical relationship between premorbidity, as quantified by aCCI, and functional outcome, assessed using path coefficients derived from SEM. Exploratory endpoints included the correlation between clinical parameters at discharge, specifically the NIHSS, and premorbidity as determined by the aCCI.

### 2.5. Statistical Analysis

All data analyses were performed by JASP (Version 0.18.3), and illustrations were created using Adobe Illustrator (Adobe, Adobe Illustrator 2023). Patients were dichotomized based on mRS scores at discharge into favorable (mRS 0–1) and unfavorable (mRS 2–6) outcomes. To simplify the analysis, we performed a median split of aCCI (aCCI 1–4 and aCCI ≥ 5).

Comparisons were undertaken using the Chi-Square test for nominal and the Mann–Whitney U Test for non-normally distributed data. Categorical variables are presented as numbers and percentages (%), quantitative variable was presented as median with interquartile range (IQR), respectively. Continuous data were presented as means with standardized differences and analyzed by Student-*t* test. To assess the complex hypothetical relationship between observed and latent variables, such as NIHSS or mRS at discharge and aCCI prior to the index event, the data were analyzed using SEM. A specific model was employed to evaluate this hypothetical relationship.

## 3. Results

Between January 2020 and December 2021, a total of 2283 patients with diagnoses corresponding to ICD-10 I63, I61, or G45 (only transient-ischemic attack—TIA—patients with stroke mimics were not included in the study) were identified in our institutional database. From this cohort, cases of acute ischemic stroke (AIS) involving the posterior circulation were isolated, yielding 514 patients. Of these, 294 were excluded from further analysis due to incomplete clinical or imaging data, a history of prior EVT, or evidence of supratentorial involvement.

To ensure the integrity of the study and minimize bias from pre-existing functional impairments, only patients with a pre-index event mRS score of 2 or less were included. Therefore, only 186 patients were included in the final analysis. The Flowchart of the patient’s selection was illustrated in Figure 1.

Over a 2-year period, 514 patients with stroke in the posterior circulation were isolated; of those, 186 remained for the final analysis.

### 3.1. Demographics and Risk Factors

The demographics, risk factors, prior therapy with anticoagulation or antiplatelet agents, and clinical and radiological presentations comparing two cohort groups, the first group with a favorable modified Rankin Scale (mRS) score of 0−1 (*n* = 111) and the second group with an unfavorable functional outcome, mRS 2−6 (*n* = 75), at discharge were presented in Table 1. Other frequencies of baseline descriptives can be found in Appendix A.

The median age for both groups was 71 years. Female patients constituted 41.4% in the first group and 34.7% in the second group. The duration of hospital stay was significantly longer in the second group, with a median value of 14 days (*p* < 0.001).

There were no significant differences between the groups regarding the distribution of risk factors or prior therapy with anticoagulation or antiplatelet agents. As expected, both NIHSS and mRS scores at presentation were higher in the second group. Additionally, brainstem involvement was notably more frequent in the second group.

### 3.2. Analysis of aCCI in Ischemic Posterior Cranial Fossa Infarcts

In our cohort, it was common to observe multiple comorbidity categories in individual patients. The most prevalent categories were diabetes and cerebrovascular disease. Notably, there were no patients diagnosed with leukemia/lymphoma, severe liver disease, or acquired immunodeficiency syndrome (AIDS). Only one patient presented with hemiparesis, and two patients with dementia had a premorbid mRS of 0–1; however, the severity of hemiparesis was not documented. The frequency distribution of each aCCI category is detailed in Table 2.

To facilitate analysis of the relationship between aCCI scores within our patient cohort, we performed a median split of the aCCI scores. The median aCCI value was 4, resulting in 114 patients (61%) with a aCCI of 0−4 and 72 patients (39%) with an aCCI of ≥5. This distribution is depicted in Figure 2. The distribution of every aCCI Index Weight was demonstrated in Appendix A.

Illustrated are the grouped patients according to aCCI stratified by a median split. Relative percentages are shown for both groups.

### 3.3. Association of aCCI with Clinical and Functional Outcome at Discharge

After utilizing a median split of the aCCI scores, we employed structural equation modeling (SEM) to explore the relationship between premorbidity scores, as measured by the aCCI prior to the index event, and functional and clinical outcomes at discharge. This analytical approach combines factor analysis and path analysis to estimate path coefficients representing direct, indirect, and total effects within our hypothesized model, as depicted in Figure 3.

The direct effect of aCCI on NIHSS is represented by the path coefficient (e.g., 0.70). The indirect effect of aCCI on NIHSS through mRS is calculated as the product of two path coefficients:○The effect of aCCI on mRS (labeled “alpha”) −1.43○The effect of mRS on NIHSS (labeled “beta”) −5.56

This represents how changes in aCCI affect NIHSS indirectly by first influencing mRS, which then affects NIHSS (0.70 + 5.56).

The total effect of aCCI on NIHSS is the sum of the direct and indirect effects. It represents the overall impact of aCCI on NIHSS, considering both direct and mediated pathways.

In path analysis, the strength of relationships between variables is indicated by path coefficients. A path coefficient of 0.1 to 0.29 indicates a small effect in the hypothetical relationship, whereas a range from 0.3 to 0.49 indicates a medium effect and 0,5 or higher a large effect, respectively.

The SEM results indicated a direct path coefficient of 0.7 for the relationship between aCCI and the NIHSS at discharge, as well as between aCCI and the overall mRS at discharge, with a path coefficient of 1.43. These coefficients highlight the impact of aCCI on patient outcomes. Additionally, we used our hypothesized model to assess the indirect effect of aCCI on NIHSS at discharge via mRS. The indirect effect was calculated as the product of two path coefficients: the effect of aCCI on mRS (1.43) and the effect of mRS on NIHSS (5.56), resulting in an indirect effect value of 7.95 (1.43 + 5.56). The total effect of aCCI on NIHSS is the sum of both direct and indirect effects, reflecting the comprehensive impact of aCCI on NIHSS through both direct influence and mediation by mRS.

In conclusion, higher aCCI scores are associated with increased scores on both mRS and NIHSS, indicating that pre-stroke comorbidities largely affect stroke severity and recovery outcomes.

## 4. Discussion

This study underscores the important influence of pre-existing comorbidities, quantified using the aCCI, on both functional and clinical outcomes following ischemic stroke. Our findings indicate that higher aCCI scores are associated with greater stroke severity, as reflected by elevated scores on the mRS and the NIHSS at discharge.

Moreover, our investigation specifically focuses on patients with posterior cranial fossa strokes, a subgroup that remains underrepresented in the current literature [22]. By examining the relationship between aCCI and outcomes in this distinct population, our study contributes novel insights into the prognostic value of comorbidity burden in a clinically challenging and often overlooked stroke subtype.

Previous research has predominantly focused on anterior circulation strokes, where the impact of comorbidities on recovery and functional outcomes is well documented [23,24]. However, our findings build on this knowledge by extending the relevance of comorbidity burden to posterior circulation strokes, demonstrating that higher aCCI scores are likewise associated with greater stroke severity and less favorable functional outcomes. By employing SEM, we were able to quantify both direct and indirect effects of comorbidities on these outcomes, providing a more detailed understanding of how pre-existing health conditions influence recovery in this specific patient population.

Finally, the findings from this study carry meaningful clinical implications for the management of patients with ischemic posterior cranial fossa strokes. The demonstrated association between higher aCCI scores and poorer outcomes emphasizes the importance of systematically assessing comorbidity burden in this population. Integrating the aCCI into routine clinical evaluation may enhance risk stratification and support more individualized treatment planning. By identifying patients at higher risk for adverse outcomes, clinicians can implement targeted, multidisciplinary interventions aimed at mitigating complications, improving functional recovery, and ultimately reducing long-term disability. These results also underscore the value of a comprehensive care model that addresses both acute stroke management and the long-term challenges of chronic disease, ensuring a more holistic approach to stroke care in this often overlooked subgroup.

### Limitations

This study has several limitations. As a retrospective, single-center analysis, it is subject to inherent biases in data completeness and consistency, given that the records were not originally collected for research purposes. The two-year study period may also limit the generalizability of findings over longer time frames. Additionally, the NIHSS, though widely used, may underestimate stroke severity in posterior cranial fossa strokes due to its limited sensitivity to region-specific deficits [22]. The absence of post-discharge follow-up data further restricts insights into long-term outcomes and recovery. Potential selection bias and unmeasured confounders, including variations in local treatment practices and the exclusion of psychosocial factors, may also influence results. These limitations highlight the need for prospective, multi-center studies with more robust data collection and broader consideration of clinical and social determinants of recovery.

## 5. Conclusions

This study highlights that in patients with ischemic posterior cranial fossa stroke, higher aCCI scores are associated with greater stroke severity at discharge and worse early functional outcomes. The findings underscore the specific influence of premorbid health status on early recovery in this patient group. Given the unique clinical challenges and prognostic uncertainty in posterior fossa strokes, future research is needed to further explore the interplay of premorbidity with other clinical, radiological, and therapeutic variables to improve outcome prediction and guide individualized care.

## Figures and Tables

**Figure 1 brainsci-15-00892-f001:**
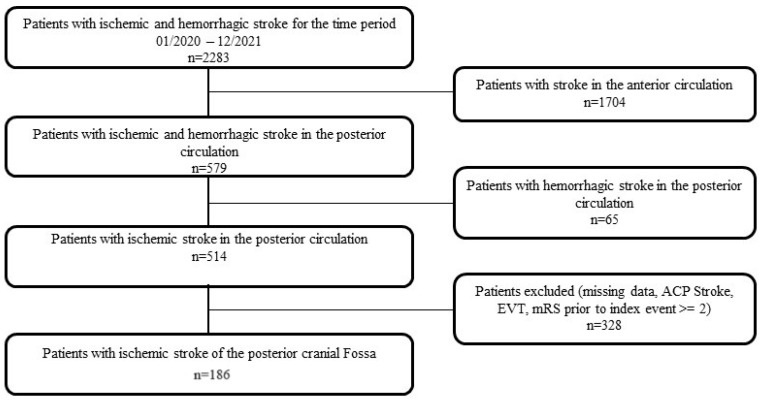
Flowchart illustrating the selection and inclusion of patient cohorts.

**Figure 2 brainsci-15-00892-f002:**
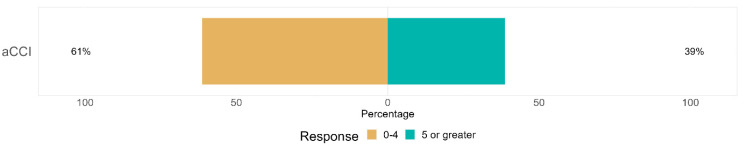
Likert plots depicting the distribution of age−adjusted Charlson Comorbidity Index (aCCI) scores in the patient cohort, stratified by median split.

**Figure 3 brainsci-15-00892-f003:**
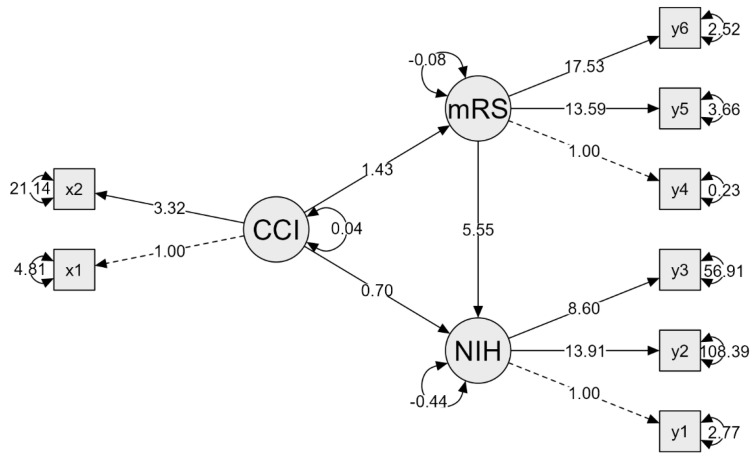
Structural equation model (SEM) demonstrating the relationship between pre−event aCCI and functional/clinical outcomes at discharge.

**Table 1 brainsci-15-00892-t001:** Frequencies and descriptive statistics of clinical and demographic characteristics in cohorts with posterior cranial fossa infarction.

Variables (*n* = 186)	Value in Total	mRS at Discharge 0−1 (*n* = 111)	mRS at Discharge ≥2 (*n* = 75)	*p* Value
Age at Indexevent *	71 (60−77)	71 (59.5−76.5)	71 (61.5−77.5)	0.427
Sex (female)	72 (38.7%)	46 (41.4%)	26 (34.7%)	0.198
Duration of hospital stay *	10 (6–14)	8 (6–12)	14 (8–24)	<0.001
Risk Factors				
● Diabetes	49 (26.3%)	28 (25.2%)	21 (28.0%)	0.163
● Previous Stroke	35 (18.8%)	21 (18.9%)	14 (18.7%)	0.769
● Atrial fibrilation	22 (11.8%)	11 (9.9%)	11 (14.7%)	0.583
● Hypertension	131 (70.4%)	80 (72.1%)	51 (68.0%)	0.865
● Overweight/Obesity (BMI > 25 mg/kg/cm^2^)	108 (58.1%)	68 (61.3%)	40 (53.3%)	0.824
● Smoking	24 (12.9%)	16 (14.4%)	8 (10.7%)	0.708
● Alkohol	13 (7.0%)	2 (1.8%)	11 (14.7%)	0.676
● Drugs	0 (0.0%)	0 (0.0%)	0 (0.0%)	-
● Hyperlipidemia	74 (39.8%)	43 (38.7%)	31 (41.3%)	0.096
Previous Therapy ● Antikoagulation				
○ Apixaban	10 (5.4%)	5 (4.5%)	5 (6.7%)	0.189
○ Rivaroxaban	4 (2.2%)	2 (1.8%)	2 (2.7%)	0.189
○ Edoxaban	6 (3.2%)	1 (0.9%)	5 (6.7%)	0.189
○ Dabigatran	1 (0.5%)	1 (0.9%)	0 (0.0%)	-
○ Enoxaparin	3 (1.6%)	3 (2.7%)	0 (0.0%)	-
○ Phenprocoumon	4 (2.2%)	2 (1.8%)	2 (2.7%)	0.868
● Anti-platelet Therapy ○ Single-Antipletelet-Therapy				
▪ Aspirin	40 (21.5%)	25 (22.5%)	15 (20.0%)	0.561
▪ Clopidogrel	7 (3.8%)	3 (2.7%)	4 (5.3%)	0.561
○ Dual antiplatelet therapy	4 (2.2%)	1 (0.9%)	3 (4.0%)	0.149
NIHSS at presentation *	1 (0−4)	1 (0−2)	4 (2−9)	<0.001
GCS at presentation *	15 (15−15)	15 (15−15)	15 (15−15)	0.001
mRS at presentation *	2 (1−3)	1 (1−2)	3 (3−4)	<0.001
Localisation of Lesion				
- Brainstem	89 (47.8%)	43 (38.7%)	46 (61.3%)	0.858
- Cerebellum	97 (60.8%)	68 (61.3%)	29 (38.7%)	0.858
LVO on Angiography	29 (15.6%)	14 (12.6%)	15 (20.0%)	0.691

Comparison of patients with favorable vs. unfavorable outcome at discharge regarding demographic data, prior medical history including risk factors and antithrombotic or antiplatelet therapy, functional and clinical scores at presentation, localization of lesion, and presence of large vessel occlusion (LVO). Values are presented either as number (percentage) or median (interquartile range). Abbreviations: BMI—body mass index; GCS- Glasgow Coma scale; mRS—modified Rankin Scale; NIHSS—National Institute of Health Stroke Scale; LVO—Large Vessel Occlusion.

**Table 2 brainsci-15-00892-t002:** Distribution of age-adjusted Charlson Comorbidity Index (aCCI) categories within the posterior cranial fossa infarct cohort.

Condition	Charlson Comorbidity Index Weight	Frequency; *n* (%)
Myocardial infarction	1	33 (17.7%)
Congestive heart failure	1	15 (8.1%)
Cerebrovascular disease	1	35 (18.8%)
Peripheral vascular disease	1	12 (6.5%)
Dementia	1	2 (1.1%)
Chronic obstructive pulmonary diseases (COPD)	1	16 (8.6%)
Connective tissue disease	1	8 (4.3%)
Peptic ulcer disease (PUD)	1	6 (3.2%)
Diabetes	1	49 (26.3%)
Mild liver disease	1	2 (1.1%)
Hemiplegia/Hemiparesis	2	1 (0.5%)
Moderate to severe renal disease	2	10 (5.4%)
Diabetes with End-organ failure	2	9 (4.8%)
Solid tumor	2	27 (14.5%)
Leukemia/Lymphoma	2	0 (0.0%)
Severe liver Disease	3	0 (0.0%)
Metastatic solid tumor	6	7 (3.8%)
Acquired immunedeficiency syndrome (AIDS)	6	0 (0.0%)

Frequencies and distribution of patients and their corresponding score in every aCCI category. Values are presented as a number (percentage). Abbreviations: aCCI—age-adjusted Charlson Comorbidity Index; COPD—chronic obstructive pulmonary disease; PUD—peptic ulcer disease; AIDS—Acquired immune deficiency syndrome.

## Data Availability

The datasets generated and/or analyzed during the current study are not publicly available due to agreements required for ethics, data protection, and privacy, but are available from the corresponding author on reasonable request.

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
