# Peer review of "Premorbid Comorbidities as Predictors of Outcome in Ischemic Posterior Fossa Stroke: A Retrospective Evaluation Using the Age-Adjusted Charlson Comorbidity Index"

_brainsci, 2025, doi:10.3390/brainsci15080892_

Round 1
Reviewer 1 Report
Comments and Suggestions for Authors
A retrospective, single-center analysis of 186 patients with posterior circulation stroke.
This group of patients did not receive endovascular treatment.
The data from these patients were evaluated to determine the correlation between the age-adjusted Charlson Comorbidity Index (aCCI) and clinical outcomes at discharge, specifically the modified Rankin Scale (mRS) and the National Institutes of Health Stroke Scale (NIHSS). Structural equation modeling (SEM) was used to elucidate the relationships between these variables.
The authors concluded that comorbidities, measured by aCCI before stroke, significantly correlated with both neurological deficit and functional outcomes at hospital discharge in patients with posterior vertebral infarction. The authors believe this indicates the high utility of aCCI as a prognostic tool for predicting outcomes in patients with posterior vertebral stroke.
The limitations of the study are detailed. The work is interesting, the static analysis is thorough, and the literature is extensive.
Notes:
1/ This raises the question of what exactly is the purpose of such an analysis? It's obvious that if a patient is highly burdened, their prognosis is worse.
2/ Can such a scale be used in situations where the indications for endovascular stroke treatment are questionable (e.g., the time of onset is long), but a high aCCI indicates a poor prognosis without treatment and can serve as an additional argument for the use of EVT?
3/ What do the authors in Table 2 understand by cerebrovascular disease? - Is a lesion in proximal part of the ICA classified in this category, or are these diseases in the intracranial segments?
4/ Why is the coexistence of AIDS such a poor prognostic factor? For example, is a patient undergoing antiviral treatment with a low rate also included in this group?
5/ please correct the number of patients - in the table and text there are 186, in the flowchart 185
Author Response
Comment 1:
A retrospective, single-center analysis of 186 patients with posterior circulation stroke.
This group of patients did not receive endovascular treatment.
The data from these patients were evaluated to determine the correlation between the age-adjusted Charlson Comorbidity Index (aCCI) and clinical outcomes at discharge, specifically the modified Rankin Scale (mRS) and the National Institutes of Health Stroke Scale (NIHSS). Structural equation modeling (SEM) was used to elucidate the relationships between these variables.
The authors concluded that comorbidities, measured by aCCI before stroke, significantly correlated with both neurological deficit and functional outcomes at hospital discharge in patients with posterior vertebral infarction. The authors believe this indicates the high utility of aCCI as a prognostic tool for predicting outcomes in patients with posterior vertebral stroke.
The limitations of the study are detailed. The work is interesting, the static analysis is thorough, and the literature is extensive.
Author's response:
We sincerely thank the reviewer for their positive and encouraging feedback on our manuscript. We are pleased that our retrospective analysis, study design, and conclusions were found to be sound and justified. We also appreciate your recognition of our efforts to integrate clinical and imaging data as well as relevant literature. Your comments are highly motivating and reinforce the clinical relevance of our findings.
Comment 2:
This raises the question of what exactly is the purpose of such an analysis? It's obvious that if a patient is highly burdened, their prognosis is worse.
Author's response:
Thank you for this insightful comment. We acknowledge that it is generally understood that a higher comorbidity burden correlates with poorer prognosis. However, currently available scores for posterior circulation strokes or stroke in general primarily assess functional outcome and stroke severity at presentation but do not adequately predict prognosis prior to therapy initiation. Our aim was to analyze how pre-existing comorbidities, as measured by aCCI, affect these commonly used scores (such as mRS and NIHSS) in this specific patient group. In future studies, we hope to further explore whether aCCI can inform therapeutic decisions regarding long-term prognosis in this population.
Comment 3
Can such a scale be used in situations where the indications for endovascular stroke treatment are questionable (e.g., the time of onset is long), but a high aCCI indicates a poor prognosis without treatment and can serve as an additional argument for the use of EVT?
Author´s response
Thank you for raising this important point. We agree that in cases where indications for endovascular therapy are uncertain, such as prolonged time from symptom onset, a tool like the aCCI could potentially aid in prognostication and decision-making, especially in posterior cranial fossa strokes with distinct clinical presentations. However, we recognize that further prospective studies are necessary to validate its utility in guiding acute therapeutic decisions.
Comment 4
What do the authors in Table 2 understand by cerebrovascular disease? - Is a lesion in proximal part of the ICA classified in this category, or are these diseases in the intracranial segments?
Author´s response
Thank you for your question regarding our definition of cerebrovascular disease. According to the specifications of the Charlson Comorbidity Index, cerebrovascular disease includes previous stroke or TIA as well as asymptomatic stenosis either in the internal carotid artery (ICA) or other intracranial segments documented in medical history.
Comment 5
Why is the coexistence of AIDS such a poor prognostic factor? For example, is a patient undergoing antiviral treatment with a low rate also included in this group?
Author´s response
We appreciate your thoughtful question on this matter. In accordance with established definitions within the Charlson Comorbidity Index, AIDS requires evidence of opportunistic infection or other manifestations of immunodeficiency. Patients with HIV infection who have low viral loads under effective antiviral therapy were not included under this category since they do not carry an equivalently poor prognosis.
Comment 6
please correct the number of patients - in the table and text there are 186, in the flowchart 185
Author´s response
Thank you very much for identifying this discrepancy. We will promptly correct all instances to ensure consistency throughout the manuscript.
Reviewer 2 Report
Comments and Suggestions for Authors
Thank you for inviting me to review this retrospective study, where authors assess the effect of premorbid comorbidities on the outcomes of the severe posterior fossa stroke. In my opinion, the manuscript is very interesting, and the methods followed are solid. I believe that the manuscript is well prepared and adds some values to the literature. I have some suggestions to improve this work:
1- In the abstract, where aCCI appears for the first time, I suggest adding a small definition to it, as its unclear.
2- Even in the introduction, CCI needs some expansion of what does it assess exactly. Not enough to mention the scoring.
3- The scoring could be moved to the methodology section instead.
4- In the methodology, the tools used need to be elaborated. The exact scoring, their exact target (severity, ...), validity,.. should be mentioned with their references.
5- The title, abstract, introduction and methods need to mention that its "ischemic" stroke. Its hard for readers to know till the flow chart in the results. Yet this should be very clear since the beginning. I also suggest focusing on inclusion/exclusion criteria in the methods.
Overall, this manuscript is interesting and very well prepared.
Good luck
Author Response
Comment 1:
Thank you for inviting me to review this retrospective study, where authors assess the effect of premorbid comorbidities on the outcomes of the severe posterior fossa stroke. In my opinion, the manuscript is very interesting, and the methods followed are solid. I believe that the manuscript is well prepared and adds some values to the literature.
Author‘s response:
We sincerely thank the reviewer for their positive and encouraging feedback on our manuscript. We are pleased that our retrospective analysis, study design, and conclusions were found to be sound and justified. We also appreciate your recognition of our efforts to integrate clinical and imaging data as well as relevant literature. Your comments are highly motivating and reinforce the clinical relevance of our findings.
Comment 2:
I have some suggestions to improve this work:
In the abstract, where aCCI appears for the first time, I suggest adding a small definition to it, as its unclear
Author‘s response:
Thank you for this insightful suggestion. We will revise the abstract to include a brief definition of the age-adjusted Charlson Comorbidity Index (aCCI) upon its first mention, ensuring clarity for a broad audience.
Comment 3:
Even in the introduction, CCI needs some expansion of what does it assess exactly. Not enough to mention the scoring.
Author‘s response:
Thank you again for your valuable comment. We will expand upon the role and components of the Charlson Comorbidity Index (CCI) in the introduction so that its purpose and significance become clearer to readers.
Comment 4:
The scoring could be moved to the methodology section instead.
Author‘s response:
We appreciate this helpful suggestion and will adjust the manuscript accordingly by relocating detailed information about scoring systems to the methodology section.
Comment 5:
In the methodology, the tools used need to be elaborated. The exact scoring, their exact target (severity, ...), validity,.. should be mentioned with their references.
Author‘s response:
Thank you for highlighting this important point. We will elaborate on all assessment tools used in our methodology section, including details regarding their scoring systems, specific targets (such as severity or functional outcome), validity, and provide appropriate references.
Comment 6
The title, abstract, introduction and methods need to mention that its "ischemic" stroke. Its hard for readers to know till the flow chart in the results. Yet this should be very clear since the beginning. I also suggest focusing on inclusion/exclusion criteria in the methods.
Author‘s response:
We appreciate your observation regarding potential confusion about stroke type. We will revise the title as well as clarify throughout the abstract, introduction, and methods sections that our study focuses specifically on ischemic stroke. Additionally, we will ensure that inclusion and exclusion criteria are clearly stated in the methods section prior to presentation of results.
Round 2
Reviewer 2 Report
Comments and Suggestions for Authors
I congratulate the authors for their revision. No further comments